# Investigation of the Effect of Induction Heating on Asphalt Binder Aging in Steel Fibers Modified Asphalt Concrete

**DOI:** 10.3390/ma12071067

**Published:** 2019-04-01

**Authors:** Hechuan Li, Jianying Yu, Shaopeng Wu, Quantao Liu, Yuanyuan Li, Yaqi Wu, Haiqin Xu

**Affiliations:** 1State Key Laboratory of Silicate Materials for Architectures, Wuhan University of Technology, Wuhan 430070, China; lihc@whut.edu.cn (H.L.); jyyu@whut.edu.cn (J.Y.); wusp@whut.edu.cn (S.W.); liyuanyuan@whut.edu.cn (Y.L.); xuhaiqin@whut.edu.cn (H.X.); 2School of Foreign languages, China University of Geosciences (Wuhan), Wuhan 430074, China; Vilcky2012@163.com

**Keywords:** asphalt concrete, induction heating, asphalt binder aging, FTIR, DSR, FCA

## Abstract

Induction heating is a valuable technology to repair asphalt concrete damage inside. However, in the process of induction heating, induced particles will release a large amount of heat to act on asphalt binder in a short time. The purpose of this paper was to study the effect of induction heating on asphalt binder aging in steel fibers modified asphalt concrete. The experiments were divided into two parts: induction heating of Dramix steel fibers coated with asphalt binder (DA) and steel wool fibers modified asphalt concrete. After induction heating, the asphalt binders in the samples were extracted for testing aging indices with Fourier Transform Infrared (FTIR), Dynamic Shear Rheometer (DSR), and Four-Components Analysis (FCA) tests. The aging of asphalt binder was analyzed identifying the change of chemical structure, the diversification of rheological properties, and the difference of component. The experiments showed that the binder inside asphalt concrete began aging during induction heating due to thermal oxygen reaction and volatilization of light components. However, there was no peak value of the carbonyl index after induction heating of ten cycles, and the carbonyl index of DA was equivalent to that of binder in asphalt concrete after three induction heating cycles, which indicated the relatively closed environment inside asphalt concrete can inhibit the occurrence of the aging reaction.

## 1. Introduction

It has been widely proven that asphalt concrete is a self-healing material and induction heating can magnify the healing ability to extend the pavement service life [1,2,3,4]. The healing mechanisms of asphalt concrete have been reported by many researchers. Bitumen is traditionally regarded as a colloidal system consisting of high molecular weight asphaltene micelles dispersed or dissolved in the lower molecular weight oily maltenes [5]. Castro and Sánchez explained the healing of asphalt mixes during rest periods by sol–gel theories [6]. Phillips proposed a three steps diffusion model to explain the healing of bitumen: (1) surface approach due to consolidating stresses and bitumen flow, (2) wetting (adhesion of two cracked surfaces to each other driven by surface energy density), and (3) diffusion and randomization of asphaltene structures. The first two steps cause the recovery of the modulus (stiffness), and the third step causes recovery of the strength [7,8]. Kringos et al. used a chemo-mechanical model to simulate healing of bitumen [9]. Garcia et al. applied capillary flow theory and flow behavior factor index to explain the healing mechanism [10,11]. All these studies provide some basis for the study of self/induction healing behavior of asphalt concrete.

The success of the test section has further accelerated the process of wide application of this technology [1,12]. Subsequently, a series of improvement studies focused on induction heating technology were conducted, among which improving induction heating efficiency became the key factor to optimize the healing ability of asphalt concrete [13,14,15,16,17,18,19]. High temperature is conducive to the occurrence and rapid progress of the healing process, which results in a higher healing rate of asphalt concrete. Liu et al. found that the optimal heating temperature was 85 °C to obtain the best healing effect for the asphalt concrete designed in his research [15]. Menozzi et al. pointed out that the total lifetime of asphalt mixture under fatigue could be increased at approximately 55 °C [19]. García et al. stated that the mechanical resistance of the test samples could be recovered up to 60% at around 100 °C, and the effect of multiple healing cycles was not compromised [20].

The above-mentioned advantages of induction heating technology, such as high induction heating efficiency, high healing rate, and multiple induction healings, emphasize the importance of temperature, especially temperature which is high enough. High temperature makes the asphalt binder to obtain better fluidity to enhance healing performance. However, it brings a significant potential damage to the asphalt material. During the process of induction heating, induced particles will produce a large amount of heat to act on asphalt binder in a short time, which may result in the possibility of thermal oxygen aging of asphalt binder. Researchers also hope that asphalt concrete can be repaired for multiple times, which greatly increases the likelihood of aging happening. Garcia et al. demonstrated there was no aging phenomenon of the binder in induction heating of asphalt mastic [18]. But Menozzi et al. [19] stated the change of flow behavior factor of the binder in asphalt mixture, which proved that the binder was aged during induction heating. These studies were not focused on the aging of asphalt binder during induction heating. Thus, there were obvious shortcomings. Garcia only studied aging of asphalt mastic, whose void volume was much smaller than that of actual pavement, and the volume of asphalt binder was too large to react with less oxygen inside asphalt mastic adequately. While Menozzi did not analyze the changes of oxygen functional groups in asphalt binders after induction heating, which is considered to be an important evidence of asphalt binder aging. In addition, their studies both concerned the aging of asphalt binders after primary induction heating, without mentioning the case after multiple induction heating. 

In view of the differences and problems in the above researches, it is necessary to redesign the experiments to study the effect of induction heating on asphalt binder aging, which will provide an important basis for the application of electromagnetic induction technology in asphalt concrete. In this paper, the experiments were divided into two parts: induction heating test of Dramix steel fibers coated with asphalt (DA) and induction heating test of steel wool fibers modified asphalt concrete. Dramix steel fiber is a tough steel fiber, and it was coated with asphalt binder film to make an induction heating binder-fiber sample, while the asphalt concrete sample was cut from the rutting plate. After induction heating test of different times under different oxygen concentration, the internal asphalt binders were extracted for testing the aging indices with Fourier Transform Infrared (FTIR) test, Dynamic Shear Rheometer (DSR) test, and Four-Components Analysis (FCA) test [21,22,23].

## 2. Materials and Experiments

### 2.1. Materials

AH-70# base asphalt obtained from Hubei Guochuang Hi-tech Material Co., Ltd., of China (Yichang, China) was used in this paper. Steel wool fibers provided by Jiangsu Golden Torch Metal products Co., Ltd (Yancheng, China) and Dramix steel fibers provided by Bekaert Corp. (Brussels, Belgium) were used as the heating units for asphalt mixtures and asphalt binders via induction heating, respectively. The morphology of the two kinds of fibers are shown in Figure 1, and the properties of bitumen, steel wool fiber, and Dramix steel fiber are shown in Table 1. The optimal content of fibers was 6% by the volume of asphalt according to previous researches [4]. Asphalt mixture with 6% steel wool fibers had the best mechanical properties (highest strength and particle loss resistance) and acceptable induction heating speed. Basalt aggregate and limestone filler were used in this study.

### 2.2. Specimen Preparation

#### 2.2.1. Dramix Steel Fiber Covered with Asphalt Binder Film (DA)

To better study the effect of high temperature produced via induction heating on asphalt binder aging, Dramix steel fibers were immersed into the heated asphalt to obtain the asphalt-fiber samples wrapped by 0.5 mm asphalt film, which were prepared for an FTIR test and FCA test, as shown in Figure 2a. For DSR test, it needed more asphalt, 3.9 g asphalt was poured into a glass dish with a diameter of 9.35 cm to obtain 0.57 mm asphalt film. The number of Dramix steel fibers was 100 to ensure adequate induction heating ability, as shown in Figure 2b.

#### 2.2.2. Asphalt Concrete Sample

AC-13 basalt asphalt concrete including steel wool fibers was designed in this paper according to Marshall design method, and the asphalt/aggregate ratio was 4.7%. The aggregate grading curves of asphalt mixtures are shown in Figure 3. In the induction heating test, the specimen was a rectangular beam cut from rutting plate in a size of 85 mm × 50 mm × 10 mm. It is important to note that in our design concept, the thinner the sample is, the better the accuracy of the test results is. There were three considerations for making the asphalt concrete specimen this size: First, in the application of induction heating, there exists gradient heating and healing, and the temperature of the surface layer is the most significant, which has been proved in previous studies [4,24], so the asphalt binders aging at the surface layer during induction heating are more representative. Second, thinner samples can eliminate as much ununiformity as possible in the binder aging of different layers due to gradient heating, which is required because we need to extract the asphalt from the mixture after induction heating. Thicker asphalt concrete sample may result in inaccurate aging testing due to an excessive gradient. Finally, the asphalt concrete gradation causes the above size to be the limit of asphalt concrete sample. Otherwise, there will be a lot of broken stones. The cut samples are shown in Figure 4.

### 2.3. Induction Heating Test

The power needed for induction heating is dependent on the conductivity and size of the specimen used for the test. Figure 5 shows the induction heating test of Dramix asphalt-fibers and asphalt concrete, corresponding to power 8.8 kW and 8.4 kW, respectively. The frequency of the induction heating apparatus was always 123 kHz. The distance between the coil and the surface of the specimen has a significant influence on the heating speed. The distance used in this paper for all samples was 10 mm according to the preliminary study [4], as shown in Figure 5. The induction heating times were 10 s for Dramix asphalt-fibers and 40 s for asphalt concrete while the corresponding temperature was 85 °C, which had proven to be the best temperature for induction healing in previous studies [15]. An infrared image camera with a resolution of 320 × 240 pixels was used to detect the temperature of the samples during induction heating, as shown in Figure 6. After each induction heating, the samples had enough time to recover to ambient temperature before starting the next induction heating test. Additionally, the phenomenon of the gradient temperature during induction heating had been demonstrated in previous studies [11,24], so in this paper, the gradient aging during induction heating was studied, and the three beams were stacked together, as shown in Figure 7. For all induction heating tests, once the surface average temperature reached 85 °C, the induction heating was stopped. For the three-layer beams induction heating test, to amplify the effect of gradient on the asphalt binder aging, the sample was heated 10 times in all, still giving the sample a full time to return to ambient temperature after each induction heating. In this paper, all Dramix steel fibers samples were induction heated 1 time. Pure asphalt (PA), as a contrast sample, was also conducted with FTIR, DSR, and FCA test. The test summary is shown in Table 2.

### 2.4. Extraction of Asphalt Binder

After induction heating, the asphalt binder was extracted by dissolution–filtration when the samples were restored to ambient temperature. For the Dramix asphalt-fibers, trichloroethylene was poured into the glass dish containing the Dramix asphalt-fibers [25]. After the asphalt coated on the Dramix steel fibers was dissolved, the fibers were removed, and the binder was stored in a closed tube for future testing. The experimental operation is shown in Figure 8. Regarding asphalt concrete, after the asphalt binder was dissolved, the solid solution mixture was filtered to separate the asphalt binder and other substances. To improve the dissolution effect and prevent the volatilization of trichloroethylene, it was necessary to dissolve the binder in a sealed state. And it is important to note in particular that the particle size of limestone filler was less than 0.075 mm. Inadequate filtration may result in the retention of limestone filler in asphalt binder, which in turn may have an impact on the test results. Therefore, the particle size of limestone fillers was determined by laser particle size analyzer (Mastersizer 2000), as shown in Figure 9. The minimum particle size of the limestone fillers was 0.363 μm, so the slow speed filter paper (pore size is 1~3 μm) was enough to filter out the most limestone filler from the asphalt mastic to obtain the pure asphalt binder. The extraction process of asphalt binder in asphalt concrete is shown in Figure 10. 

### 2.5. Fourier Transform Infrared (FTIR) Test

The chemical structures of the virgin asphalt and extractive asphalt binders before and after induction heating were studied by a Fourier transform infrared (FTIR, Nexus, ThermoNicolet Crop., Waltham, MA, USA). In this paper, FTIR tests were performed under a scanned area between 4000 and 400 cm^−1^. And the scanning resolution was 4 cm^−1^. The carbonyl functions C=O (centered around 1700 cm^−1^) were monitored by studying their changes in spectra. The carbonyl functions C=O can provide the information about oxidation of asphalt. The carbonyl functions C=O index could be calculated by the area of its bands by the following equation [26]:
(1)$$I_{C=O}=\frac{\mathrm{Area}\;\mathrm{of}\;\mathrm{carbonyel}\;\mathrm{band}\;\mathrm{centered}\;\mathrm{around}\;1700{\;\mathrm{cm}}^{-1}}{\sum \;\mathrm{Area}\;\mathrm{of}\;\mathrm{spectral}\;\mathrm{bands}\;\mathrm{between}\;2000\;\mathrm{and}\;600{\;\mathrm{cm}}^{-1}}$$

### 2.6. Dynamic Shear Rheometer (DSR) Test

DSR (MCR101, Anton Paar, Graz, Austria) was applied to investigate the rheological properties of extracted asphalt binder previously. Frequency sweep test was performed in the range of 40 to 76 °C. Seven temperature levels were measured at intervals of 6 °C. At each temperature level, the frequency scanning range was 0.1~100 Hz. To ensure the uniform temperature of the specimen, the temperature equilibrium time of the asphalt specimen at each sweeping temperature was 3 min. A plate with 25 mm diameter and 1 mm gap were used. According to Liu et al.’s research [27], the master curve was simulated through the logPen. model.

### 2.7. Four-Component Analysis (FCA) Test

Asphalt consists of various molecular weights of hydrocarbons and their derivatives and based on the relative molecular size and polarity of asphalt, it can be divided into four components, namely saturates, aromatics, resins, and asphaltenes. To test the effects of induction heating on these four components, asphalt binders induced heated in different conditions and times were tested through TLC-FID (Iatron Laboratories Inc., Tokyo, Japan). According to [23], two percent (*w*/*v*) solutions of asphalt binders were prepared in dichloromethane, and 1 µL sample solution was spotted on chromarods. There was a three-stage process for the separation of asphalt fractions. The first stage was in n-heptane (70 mL) and expanded to 100 mm of the chromarods, the second stage in toluene/n-heptane (70 mL, 4/1 by volume) was developed to 50 mm of the chromarods, and the last development was in toluene/ethanol (70 mL, 11/9 by volume) and expanded to 25 mm of the chromarods. The solvent was dried in an oven at 80 °C after each stage. Then, the chromarods were scanned in the TLC-FID analyzer. Four chromarods were tested for each sample, and finally, the average values were used as the results. In this paper, 10 parallel tests were performed on each sample.

## 3. Results and Discussion

### 3.1. DA Induction Heating

Figure 11 shows the infrared images of Dramix steel fibers and steel wool fibers after induction heating for 10 s and 5 s, respectively. It was found that the maximum temperature reached to 142 °C and 87.1 °C, with a heating rate 8.5 °C/s and 12.2 °C/s. Predictably, if the heating time is prolonged, the temperature of the fibers will be higher. Because of the rapid temperature rising of asphalt wrapped on the surface of fibers, asphalt binder is subjected to very high thermal radiation directly, which is enough to cause the asphalt binder aging.

Dramix asphalt-fibers were completely exposed to air during induction heating. In this condition, the amount of oxygen was sufficient for binder aging, which could rule out that there would be no binder aging in asphalt concrete due to lack of oxygen. The purpose of the experiment was to verify whether asphalt binders would age in the case of sufficient oxygen during induction heating.

### 3.2. Rheological Properties Analysis

The master curves of complex moduli and phase angles of asphalt binders after induction heating in different condition are shown in Figure 12 and Figure 13, respectively. The reference temperature for establishing the master curves was 46 °C. From Figure 12 and Figure 13, the complex moduli and phase angle of all asphalt binders changed obviously after induction heating, and the change was aggravated with the increase of induction heating times. After induction heating, the complex moduli and phase angle of asphalt binders increased and decreased severally. Compared with the complex moduli, the change and trend of phase angle were more irregular, but it still proved that the asphalt binder was aging during induction heating. The increase tendency of complex moduli indicated that the resistance of asphalt to deformation under repeated shear loading increased. Meanwhile, the decrease tendency of phase angle demonstrated that the ratio of elastic modulus (or storage modulus) to complex modulus increased after induction heating. These were the aging characteristics of asphalt binders. These changes can be explained in two aspects: On the one hand [28], some asphalt molecules produced oxygen-containing functional groups with higher molecular weights as a result of the oxidation reaction. On the other hand [29], due to the volatilization of the light components in the asphalt at high temperature, the proportion of the light component fraction decreased, while the proportion of the weight component fraction increased. This imbalance of components can lead to the deterioration of physical and rheological properties of asphalt binders, which will be validated and discussed according to FTIR test and FCA test in the following two chapters.

### 3.3. Chemical Structure Analysis

#### 3.3.1. Multiple Induction Heating

Figure 14 shows the changing areas of carbonyl functions and Figure 15 presents the changes in FTIR index of asphalt binders obtained from different induction heating conditions. From Figure 14, it can be observed that with the increase of induction heating times, the carbonyl index in asphalt binder increased gradually, but there was no peak value. For pure asphalt, *I*_C=O_ was 0.000531, which was far below the value of other samples, indicating that slow aging occurred during production and storage. After induction heating for 10 cycles, *I*_C=O_ reached to 0.006231, which demonstrated that the asphalt binder aging was very obvious at this time. However, for DA, its carbonyl index was equivalent to the value of the asphalt binder inside the asphalt concrete after inducting heating for three cycles, which was more serious than that after induction heating for one and two cycles. This showed that in the case of more oxygen, the asphalt binder did incur more serious thermal oxygen aging during induction heating, and in the meantime, the relatively closed space in asphalt concrete could restrain the aging.

#### 3.3.2. Gradient Heating of Three-Layer Beams

Figure 16 shows the I_C=O_ index of asphalt binder after induction heating in different layers. The *I*_C=O_ of the one, two and three layers were 0.00619, 0.00516, and 0.00384 in turn, which was the same gradient phenomenon with the induction heating temperature (40 s, 8.6 kW) in Figure 6. The carbonyl index of asphalt binder in the surface layer was similar to that in asphalt concrete which was induced heated for 10 cycles in the preceding section. With the decrease of induction temperature, the carbonyl functional groups produced inside the asphalt binder gradually decreased, indicating that the gradient aging of binder happened during induction heating. The surface average heating temperature was 85 °C, which was not enough to cause the asphalt aging in theory, but the actual situation was that even if the average temperature of the third layer was only 47.3 °C, the asphalt binder was still aged. This was easy to understand, because the so-called average temperature was the average temperature of asphalt concrete, and from the above study, it was found that steel fibers gave off a lot of heat in a short period during induction heating, which was enough for the asphalt binders aging. From FTIR tests, it was found that one of the reasons for the asphalt binders aging was the thermal oxygen aging reaction of asphalt molecules with oxygen at high temperature, and the other mechanism of asphalt aging will be explained in the next chapter.

### 3.4. Four-Component Analysis

Figure 17 shows the component fractions of asphalt binders after induction heating in different conditions. Significant changes of four component fractions of different asphalt binders took place. After induction heating for 10 cycles, the fractions of saturates decreased from 20.29% to 13.69%, and the fractions of aromatic decreased from 37.53% to 30.16%, while the fractions of resins increased from 29.08% to 35.82%, and the fractions of asphaltenes increased from 13.10% to 20.33%. From Figure 17, the trend of components changes showed that the number of induction heating cycles was the decisive factor. The change of component fractions may be due to volatilization of light components (saturates and aromatic) or transition of light components to weight components at high temperature. In addition, the component fractions of DA were similar to that of the asphalt binder after induction heating for 3 cycles. This may be because open space was more conducive to the volatilization of light components, thus, avoiding the reattachment of volatile light components to the interior of asphalt concrete.

## 4. Conclusions

In this research, the effect of induction heating on asphalt binder aging in steel fibers modified asphalt concrete was investigated. Based on the results discussed above, the following conclusions could be drawn:
It was demonstrated that the asphalt binder inside asphalt concrete began aging during induction heating due to the rapid temperature rise of asphalt wrapped on the surface of fibers, whose aging mechanisms were thermal oxygen aging and volatilization of light components or transition of light components to weight components.According to DSR, the complex moduli and phase angle of asphalt binders increased and decreased severally after induction heating, indicating that the rheological properties of asphalt binders changed.For the binder inside asphalt concrete, there was no peak value of carbonyl index after ten cycles of induction heating, and the carbonyl index of DA was equivalent to that of asphalt binder after three cycles induction heating, indicating that relatively closed environment inside the asphalt concrete could restrain the thermal oxygen aging.The number of induction heating was the decisive factor to influence the change of asphalt binder component fractions, and the binder component fractions changed more slowly compared to DA.Although the asphalt binder aging inside asphalt concrete was slower, it was still necessary to study the effect of binders aging on the healing performance of asphalt concrete.

**Further research:** The test will be conducted in two aspects: On the one hand, the change in healing performance of the asphalt binder itself will be investigated, and we plan to evaluate it by the flow behavior factor index, which is the most recognized method now. On the other hand, we will design experiments to verify whether the incomplete strength recovery of asphalt mixture after induction healing is related to the aging of asphalt binder.

## Figures and Tables

**Figure 1 materials-12-01067-f001:**
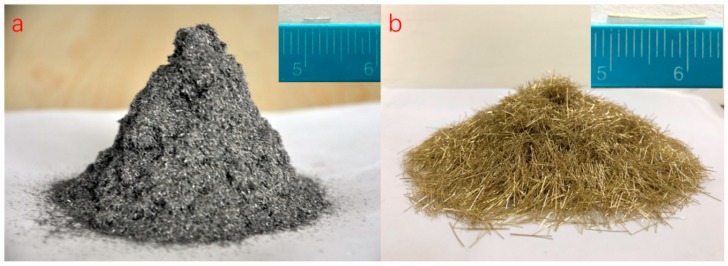
Morphology of steel wool fibers (**a**) and Dramix steel fibers (**b**).

**Figure 2 materials-12-01067-f002:**
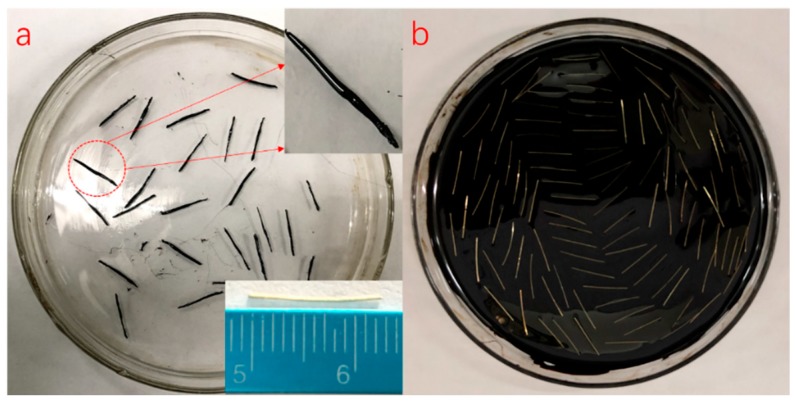
Dramix steel fibers coated with asphalt binder (DA) for (**a**) Fourier Transform Infrared (FTIR) and Four-Components Analysis (FCA) tests and (**b**) Dynamic Shear Rheometer (DSR) test.

**Figure 3 materials-12-01067-f003:**
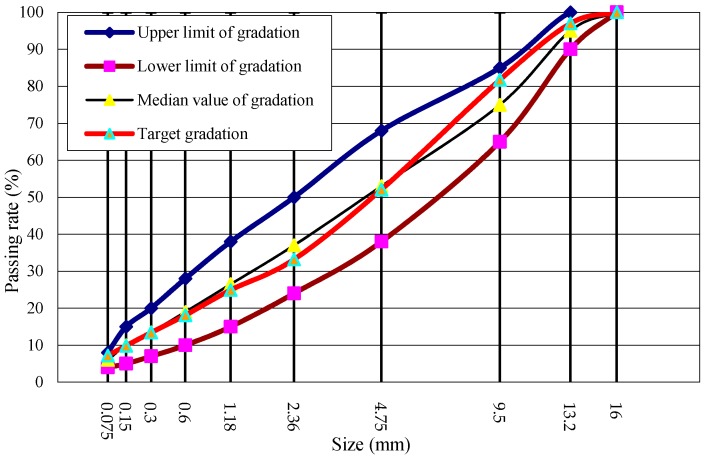
Grading curve of AC-13 steel wool fibers modified basalt asphalt mixture.

**Figure 4 materials-12-01067-f004:**
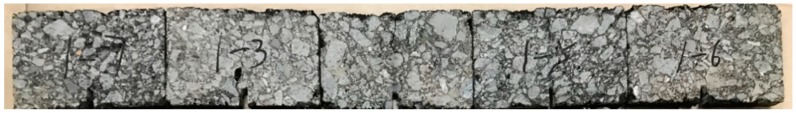
Induction heating test samples.

**Figure 5 materials-12-01067-f005:**
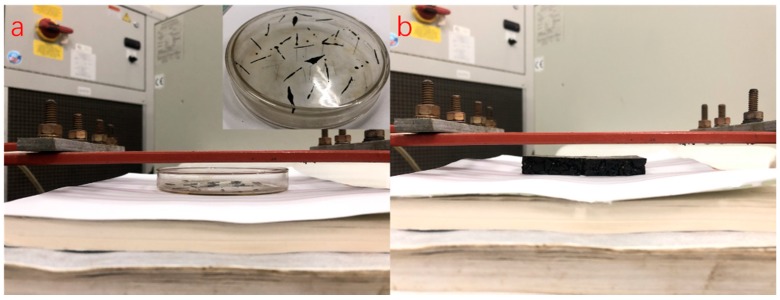
Induction heating test of Dramix asphalt-fibers (**a**) and asphalt concrete (**b**).

**Figure 6 materials-12-01067-f006:**
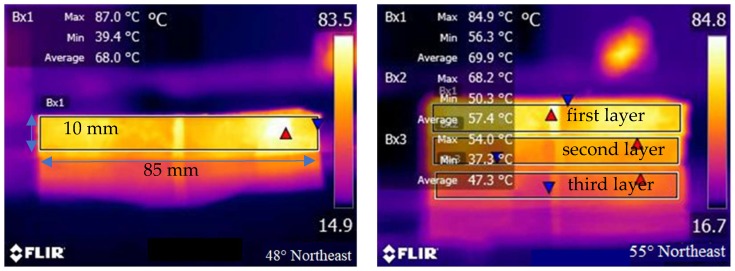
Infrared imaging of samples during induction heating.

**Figure 7 materials-12-01067-f007:**
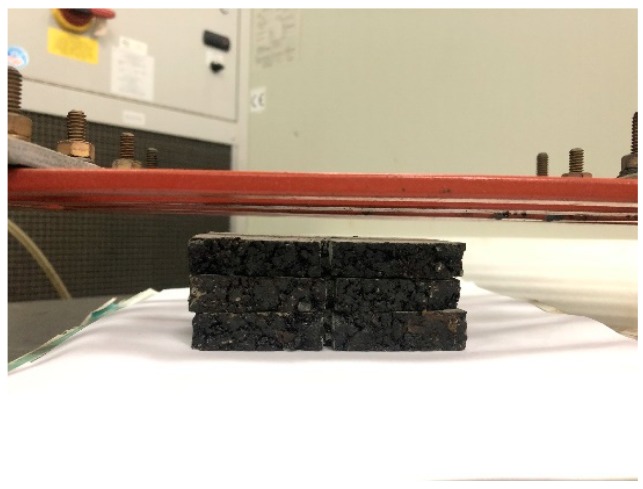
Induction heating of three-layer beams.

**Figure 8 materials-12-01067-f008:**
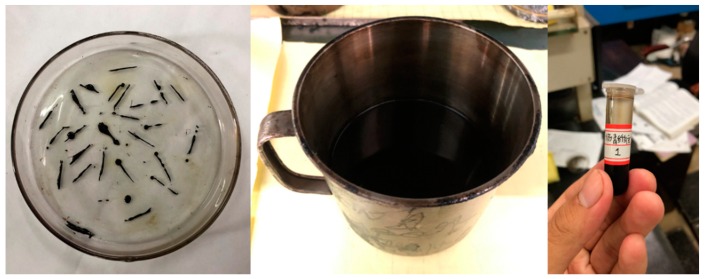
Extraction and storage of asphalt binder coated on Dramix steel fibers.

**Figure 9 materials-12-01067-f009:**
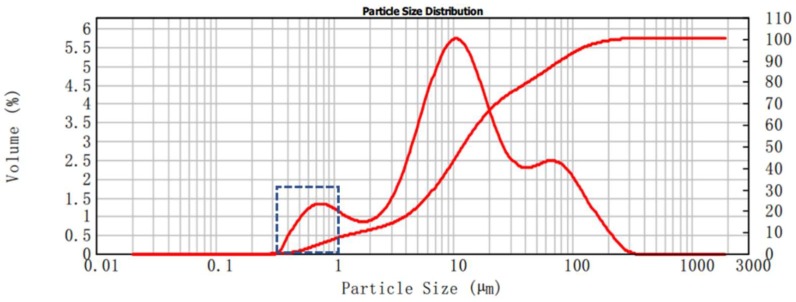
Laser particle size analysis of limestone fillers.

**Figure 10 materials-12-01067-f010:**
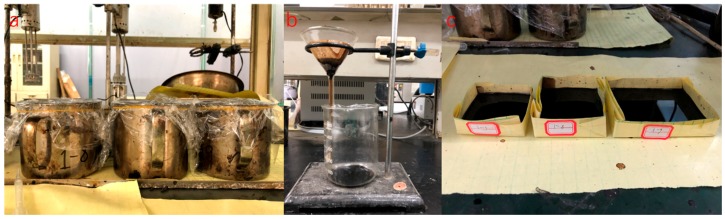
Extraction process of asphalt binder in asphalt concrete: dissolution (**a**), filtration (**b**), and volatilization (**c**).

**Figure 11 materials-12-01067-f011:**
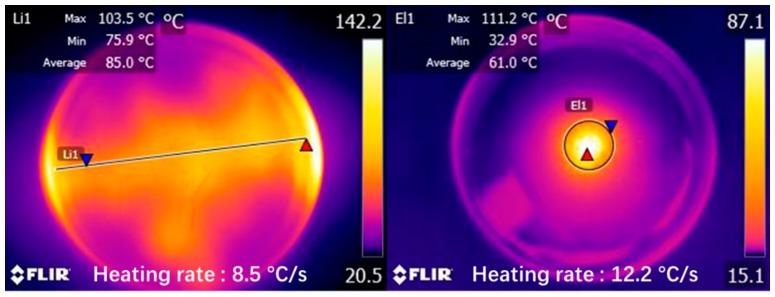
Infrared images of Dramix steel fibers and steel wool fibers after induction heating.

**Figure 12 materials-12-01067-f012:**
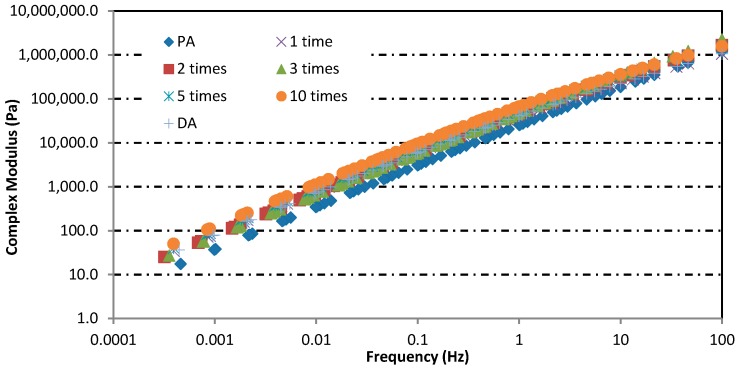
Complex modulus master curve of asphalt binders in different induction heating conditions.

**Figure 13 materials-12-01067-f013:**
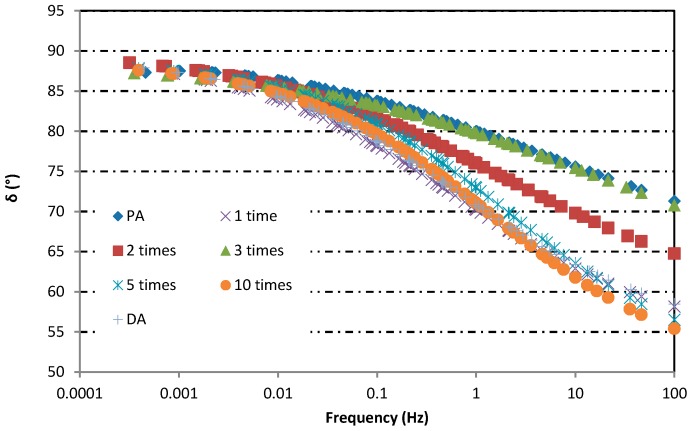
Phase angle master curve of asphalt binders in different induction heating conditions.

**Figure 14 materials-12-01067-f014:**
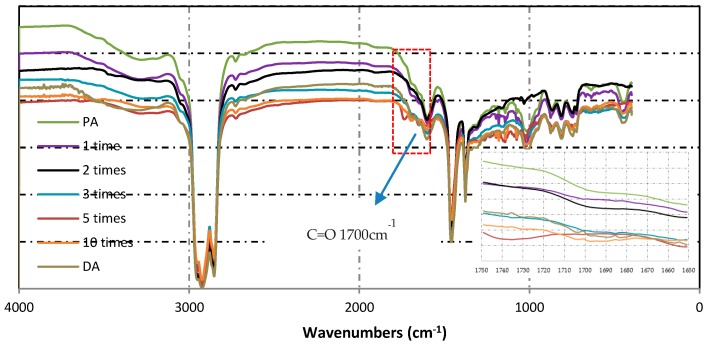
FTIR spectra of asphalt binders before and after induction heating.

**Figure 15 materials-12-01067-f015:**
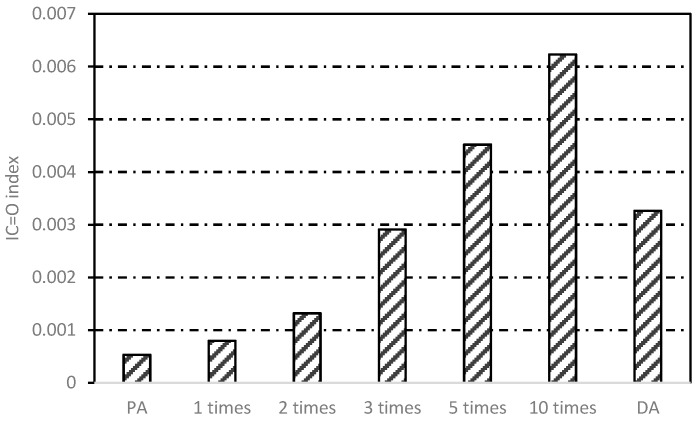
*I*_C=O_ variation of asphalt binders before and after induction heating.

**Figure 16 materials-12-01067-f016:**
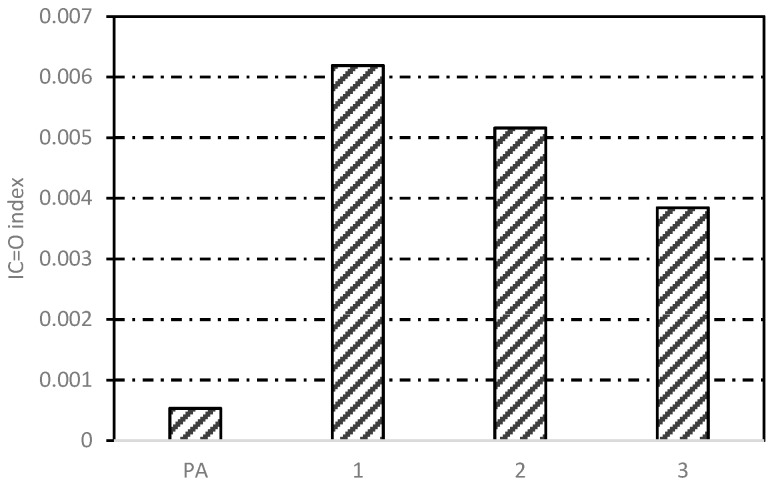
*I*_C=O_ index of asphalt binder after induction heating in different layers.

**Figure 17 materials-12-01067-f017:**
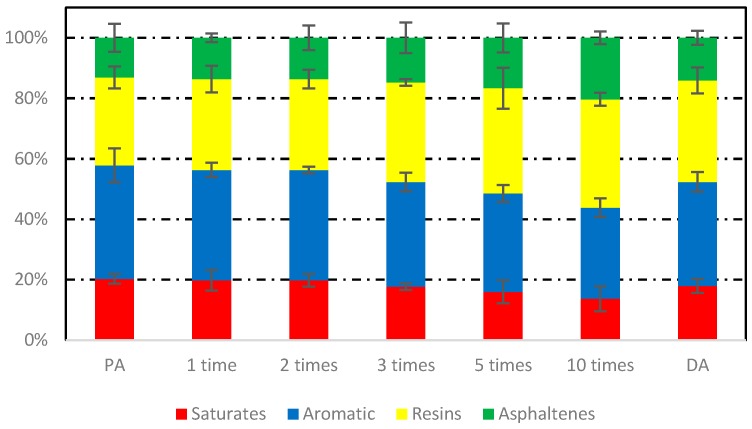
Component fractions of asphalt binders before and after induction heating in different conditions.

**Table 1 materials-12-01067-t001:** The properties of bitumen and steel fibers.

Materials	Properties	Values	Specifications
Bitumen	Penetration (25 °C, 100 g, 5 s, 0.1 mm)	68	60–80
	Ductility (15 °C, cm)	>100	100
	Softening point (°C)	47.5	47
	Density (g/cm^3^)	1.034	-
Steel wool fiber	Average length (mm)	4.2	-
	Equivalent diameter (μm)	70–130	-
	Density (g/cm^3^)	7.8	-
	Average heating rate (°C/s)	12.2	
Dramix steel fiber	Average length (mm)	13	-
	Equivalent diameter (μm)	200	-
	Density (g/cm^3^)	7.8	-
	Minimum tensile strength (N/mm^2^)	2	-
	Average heating rate (°C/s)	8.5	

**Table 2 materials-12-01067-t002:** The test summary.

Specimen	Number	Induction Heating Times	Test
PA	–	0	FTIR, DSR & FCA
Single-layer beam	1	1	
	1	2	
	1	3	
	1	5	
	1	10	
DA	25 + 100	1	
Three-layer beams	1	10	FTIR

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
