# Peer review of "Investigation of the Effect of Induction Heating on Asphalt Binder Aging in Steel Fibers Modified Asphalt Concrete"

_materials, 2019, doi:10.3390/ma12071067_

Reviewer 1 Report

The development of self-healing material is a promising research path because of the potential savings that could be made regarding inspection maintenance and reparation works. Some solutions have been engineered in many industries (coatings, polymers, civil engineering) but the durability of self-healing materials is rarely investigated. Self-healing asphalt has been introduced for around 10 years but there are only few in situ applications and some work is still needed before a large scale usage.

This manuscript exposes a study on the effect of induction heating on asphalt binder aging in asphalt concrete, in the context of induction heating-based self-healing asphalt development. Steel fibers coated with asphalt binder and asphalt concrete are studied using dynamic shear rheometer (DSR), Fourier transform infrared test (FTIR), four-component analysis and infrared imaging in inder to evaluate the influence of repeated heating cycles (as it would occur in the case of multiple healing periods).

Globally the manuscript is well presented and present new pieces of information that would be of interest to the readers of the journal. The introduction is good and provides a clear and concise background emphasizing the interest of the study. Then the section presenting the materials and experiments is rather good although some minor remarks should be addressed. The results are clearly presented and briefly discussed. Finally the conclusions are supported by the results.

Some minor remarks may be addressed before publication:

- l 86: remove ‘the’ before ‘basalt’

- figure 1: please add a scale indication (could be a common day-to-day life object)

- l 95: can you provide an estimation of the thickness of asphalt around the fibers?

- figure 2: scalebar again

- l 103 – 116: It is necessary to explain how many samples were studied. Because of the small size of the samples, is there a greater measurements dispersion? The aggregate size should also be mentioned so that the reader can estimate whether the small specimens are representative volumes or not.

- section 2.3: please sum up the tests and the specimens denominations in one table for example (number of heating times, DA, PA)

- figure 6: some direction indications and scales are welcomed

- fig 9: the table under the figure is unnecessary in my opinion

- l 208: you mention 2 heating rates. It may be clarified in the materials and methods section

- fig 13 to 17: please remove the green background

- fig 14: a zoom around 1700 cm-1 may be useful as a ‘b)’ figure

- l 283: remove ‘the’ before ‘significant changes’

- table 2 and figure 17 represent the same results, thus table 2 may be deleted.

- Figure 17: error bars are not correctly visible

- ref 25 is just a link

Author Response

Reviewer #1:

The authors want to thank the reviewer for its thorough work, acknowledging that the paper has been greatly improved by its input.

The development of self-healing material is a promising research path because of the potential savings that could be made regarding inspection maintenance and reparation works. Some solutions have been engineered in many industries (coatings, polymers, civil engineering) but the durability of self-healing materials is rarely investigated. Self-healing asphalt has been introduced for around 10 years but there are only few in situ applications and some work is still needed before a large scale usage.

This manuscript exposes a study on the effect of induction heating on asphalt binder aging in asphalt concrete, in the context of induction heating-based self-healing asphalt development. Steel fibers coated with asphalt binder and asphalt concrete are studied using dynamic shear rheometer (DSR), Fourier transform infrared test (FTIR), four-component analysis and infrared imaging in inder to evaluate the influence of repeated heating cycles (as it would occur in the case of multiple healing periods).

Globally the manuscript is well presented and present new pieces of information that would be of interest to the readers of the journal. The introduction is good and provides a clear and concise background emphasizing the interest of the study. Then the section presenting the materials and experiments is rather good although some minor remarks should be addressed. The results are clearly presented and briefly discussed. Finally the conclusions are supported by the results.

Some minor remarks may be addressed before publication:

- l 86: remove ‘the’ before ‘basalt’

Done.

- figure 1: please add a scale indication (could be a common day-to-day life object)

Done.

- l 95: can you provide an estimation of the thickness of asphalt around the fibers?

Sure. By calculating the weight of steel fiber before and after coating asphalt, the thickness of the film is estimated to be about 0.5mm.

- figure 2: scalebar again

Done.

- l 103 – 116: It is necessary to explain how many samples were studied. Because of the small size of the samples, is there a greater measurements dispersion? The aggregate size should also be mentioned so that the reader can estimate whether the small specimens are representative volumes or not.

Now the number of every kind of specimen was summed up in table 2 in section 2.3. For the aggregate size, please refer to Figure 3.

- section 2.3: please sum up the tests and the specimens denominations in one table for example (number of heating times, DA, PA)

Done. Please refer to table 2.

- figure 6: some direction indications and scales are welcomed

Done.

- fig 9: the table under the figure is unnecessary in my opinion

The table has been removed.

- l 208: you mention 2 heating rates. It may be clarified in the materials and methods section

Done. We added the data to Table 1 so that the reader could understand it better.

- fig 13 to 17: please remove the green background

Done. Our original manuscript didn't have that background. Whatever, the errors were corrected.

- fig 14: a zoom around 1700 cm-1 may be useful as a ‘b)’ figure

Done.

- l 283: remove ‘the’ before ‘significant changes’

Done.

- table 2 and figure 17 represent the same results, thus table 2 may be deleted.

Done.

- Figure 17: error bars are not correctly visible

The problem has been solved.

- ref 25 is just a link

This paper is closely related to this study, and it has already been online. But for now, it can only be referenced through doi.

Reviewer 2 Report

The paper is aimed at experimentally reporting the effect of induction heating on asphalt binder aging in asphalt concrete. Healing mechanisms are analyzed under induction heating. The work contains potentially good material to be published in Materials.

In the Reviewer's opinion the paper needs some improvements before the approval.

The following recommendations/clarifications should be considered:

·       Title: I would suggest to modify it to highlight that the paper deals with Dramix and wool steel fibers;

·       In section 1, after the State of the Art, the Authors should better clarify what are the key novelties of this paper and the main contributions of this work beyond the current SoA. They are missing or only properly reported.

·       Apparently, all the reviewed and analyzed works, discussed in the introduction section, does not have fibers to improve the induction heating mechanisms (only the work [19] studied the effect of conductive fibers). The authors need to highlight and prove that the use of fiber reinforcements are really necessary and worth of investigation in this matter.

·       The % amount of fibers is very high (6% in volume is a huge amount). Normally, for this value the concrete casting is very difficult and particular strategy must be implement to avoid such problems (i.e. using of special additives). How did the authors solve this?

·       The authors reported interesting values to check the repairing of asphalt concrete due the induction heating technology. However, no mechanical tests were performed for this purpose. Perhaps could be also valuable to realize it.

·       Please add the Table of the mix design per each mixture.

·       Was the heating rate affecting the results? Please discuss this point.

·       The authors wrote that after induction heating, the asphalt binder was extracted by dissolution-filtration when the temperature of samples was restored to ambient temperature. The Reviewer is wondering that this process could affect the material properties of the analyzed samples.

·       Future developments which will follow to this research paper should be written. It should be reported at the end of the concluding section. Maybe envisioning them to perform more experimental activities to check, also mechanically, the healing effects produced in the analyzed materials.

Author Response

Reviewer #2:

The authors want to thank the reviewer for its thorough work, acknowledging that the paper has been greatly improved by its input.

The paper is aimed at experimentally reporting the effect of induction heating on asphalt binder aging in asphalt concrete. Healing mechanisms are analyzed under induction heating. The work contains potentially good material to be published in Materials.

In the Reviewer's opinion the paper needs some improvements before the approval.

The following recommendations/clarifications should be considered:

·       Title: I would suggest to modify it to highlight that the paper deals with Dramix and wool steel fibers;

Done.

·       In section 1, after the State of the Art, the Authors should better clarify what are the key novelties of this paper and the main contributions of this work beyond the current SoA. They are missing or only properly reported.

Thanks for your advice. At present, it is rare to study the effect of induction heating on asphalt binder aging in steel fibers modified asphalt concrete. The relevant references that the author can consult has been mentioned in the article. Please refer to: [Álvaro García, Schlangen E, Ven M V D, et al. Induction heating of mastic containing conductive fibers and fillers[J]. Materials & Structures, 2011, 44(2):499-508.] & [Menozzi A, Garcia A, Partl M N, et al. Induction healing of fatigue damage in asphalt test samples[J]. Construction & Building Materials, 2015, 74:162-168.] The innovation of this paper is to design the test and prove the asphalt binder aging in steel fibers modified asphalt concrete during induction heating, at the same time, it quantifies the aging degree. The description has been presented in the last paragraph of the introduction.

·       Apparently, all the reviewed and analyzed works, discussed in the introduction section, does not have fibers to improve the induction heating mechanisms (only the work [19] studied the effect of conductive fibers). The authors need to highlight and prove that the use of fiber reinforcements are really necessary and worth of investigation in this matter.

Thanks for your advice. From the point of view of the title of the article, it is true that only reference 19 refers to steel fibers. In fact, the references cited in this paper, which involves healing or self-healing or induction healing, are all about the electromagnetic inducting healing behavior of steel fibers modified asphalt concrete. At present, the research on self-healing of asphalt concrete involves several aspects, such as chemical healing (microcapsule/capsule healing & healing agent), physical healing (electromagnetic induction method & microwave method & Infrared method). Generally speaking, induction heating/healing means steel fibers modified asphalt concrete. In this study, in order to give a clearer indication of the subject, we made a special note in the title “steel fibers modified asphalt concrete”.

·       The % amount of fibers is very high (6% in volume is a huge amount). Normally, for this value the concrete casting is very difficult and particular strategy must be implement to avoid such problems (i.e. using of special additives). How did the authors solve this?

The first thing I want to explain is that the fiber is 6% of the asphalt binder volume. Because it's possible to think of 6% as 6% of the volume of asphalt concrete, which leads to a misunderstanding. If you know that 6% is 6% of the volume of asphalt, in order to avoid the agglomeration of steel fibers, mixing sequence and method are key. The related problems have been well solved in previous studies. Please refer to: [Liu Q , Schlangen E , álvaro García, et al. Induction heating of electrically conductive porous asphalt concrete[J]. Construction & Building Materials, 2010, 24(7):1207-1213.], [Liu Q , Yu W , Schlangen E , et al. Unravelling porous asphalt concrete with induction heating[J]. Construction and Building Materials, 2014, 71:152-157.] & [Yu, W. Study of the Induction Healing Behaviors of Hot Warm Mix Asphalt. Master’s Thesis, Wuhan

University of Technology, Wuhan, China, 2017].

·       The authors reported interesting values to check the repairing of asphalt concrete due the induction heating technology. However, no mechanical tests were performed for this purpose. Perhaps could be also valuable to realize it.

Your advice is very meaningful. In fact, a lot of work has been done on the mechanical properties of induction healing asphalt concrete. Please refer to: [Liu Q, Chen C, Li B, et al. Heating Characteristics and Induced Healing Efficiencies of Asphalt Mixture via Induction and Microwave Heating[J]. Materials, 2018, 11(6).] & [https://doi.org/10.1016/j.conbuildmat.2019.03.052]. However, it is new and meaningful to evaluate the change of mechanical properties of induction healing asphalt concrete from the point of view of asphalt binder aging. We are currently experimenting with this, and this part of the description has been added to “future research” in this paper.

·       Please add the Table of the mix design per each mixture.

Done. Please refer to table 2.

·       Was the heating rate affecting the results? Please discuss this point.

Thanks for your question. There is no doubt that the heating rate will affect the test results. But as far as this article is concerned, I don't think the two steel fibers chosen will affect the results. Because the heating rates of these two kinds of steel fibers are 8.5 °C/s and 12.2 °C/s. It is clear that the aging of asphalt binder is due to the large amount of heat produced by the inductive additives (steel fibers in this study) in a short period of time acting on the asphalt binder, and the heating rates of the two kinds of steel fibers selected in this paper are high enough to avoid the impact of different materials. In fact, it's best to choose the same kind of steel fiber. However, in the aging study of asphalt film, the size of steel wool fiber is too small for sample preparation, so we choose the Dramix steel fiber with similar heating rate to replace it. In addition, we select the index, different induction heating time to reach the same temperature (85 °C in this study), to evaluate the exothermic ability of the two kinds of steel fibers, which is the key factor to impact the aging of asphalt binder. This evaluation was described in the previous section “induction healing test”.

·       The authors wrote that after induction heating, the asphalt binder was extracted by dissolution-filtration when the temperature of samples was restored to ambient temperature. The Reviewer is wondering that this process could affect the material properties of the analyzed samples.

Thanks for your good question. For the study in this paper, the asphalt binder aging is mainly related to the chemical process. Dissolution-filtration is a physical process and therefore does not affect the properties of the material. You may worry that the solvent will affect the properties of the material, in fact, organic solvents such as trichloroethylene or carbon disulfide are very common in the dissolution of asphalt binder, they don't change the chemical properties of asphalt binders. In order to reassure possible concerns of the reviewer and the readers, we have added reference. Please refer to: [Zeng W, Wu S, Wen J, et al. The temperature effects in aging index of asphalt during UV aging process[J]. Construction & Building Materials, 2015, 93:1125-1131.]

·       Future developments which will follow to this research paper should be written. It should be reported at the end of the concluding section. Maybe envisioning them to perform more experimental activities to check, also mechanically, the healing effects produced in the analyzed materials.

Thanks for your good advice. Future research has been added after conclusion. In fact, we've already done some research. The test will be conducted in two aspects: on the one hand, the change in healing performance of the asphalt binder itself will be investigated, and we plan to evaluate it by the flow behavior factor index, which is the most recognized method now. On the other hand, we will design experiments to verify whether the incomplete recovery of asphalt mixture strength after induction healing is related to the aging of asphalt binder.

Reviewer 3 Report

The paper is interesting, well-conceived and well structured. The reviewer, would recommend the authors to thoroughly revise the paper in terms of English and sentence structure.

Author Response

Reviewer #3:

The authors want to thank the reviewer for its thorough work, acknowledging that the paper has been greatly improved by its input.

The paper is interesting, well-conceived and well structured. The reviewer, would recommend the authors to thoroughly revise the paper in terms of English and sentence structure.

Thanks for your advice. We have invited native English speaker to check this article, and the corresponding changes have been noted in the article.

Round  2

Reviewer 2 Report

The paper can be accepted